# Causally Fair Node Classification on Non-IID Graph Data

**Yucong Dai**                                                                          *yucongd@clemson.edu*
*Clemson University*

**Lu Zhang**                                                                                *lz006@uark.edu*
*University of Arkansas*

**Yaowei Hu**                                                                     *yaowei.hu@walmart.com*
*Walmart Inc.*

**Susan Gauch**                                                                          *sgauch@uark.edu*
*University of Arkansas*

**Yongkai Wu**                                                                       *yongkaw@clemson.edu*
*Clemson University*

**Reviewed on OpenReview:** *https://openreview.net/forum?id=AwptwzGld5*

## Abstract

Fair machine learning seeks to identify and mitigate biases in predictions against unfavorable populations characterized by demographic attributes, such as race and gender. Recent research has extended fairness to graph data, such as social networks, but many studies neglect the causal relationships among data instances. This paper addresses a prevalent challenge in many fair machine learning research, which typically assumes independent and identically distributed (IID) data, from the causal perspective. Specifically, this work targets the circumstance where nodes with different neighborhood structures follow different causal mechanisms, violating the invariance assumptions required for classical structural causal models and *do*-calculus. We base our research on the Network Structural Causal Model (NSCM) framework and develop a Message Passing Variational Autoencoder for Causal Inference (MPVA) to compute interventional distributions for causally fair node classification. We establish theoretical soundness under two conditions: Decomposability and Graph Independence. These conditions formalize when causal mechanism heterogeneity can be overcome by constructing a structural representation that restores invariance and facilitates the computation of interventional distributions using *do*-calculus in non-IID settings. Empirical evaluations on semi-synthetic and real-world datasets demonstrate that MPVA outperforms conventional methods by effectively approximating interventional distributions and mitigating bias. Our findings demonstrate the potential of causality-based fairness in complex ML applications and motivate future work on relaxing the classic assumptions in algorithmic fairness.

## 1 Introduction

As machine learning systems become increasingly common in real-world decision making, identifying and mitigating prediction bias remains essential for equity and reliability in high-stakes settings (Caton & Haas, 2020; Zafar et al., 2017a;b; Mehrabi et al., 2021; Pessach & Shmueli, 2023; Zliobaite, 2017; Quy et al., 2022; Wan et al., 2022). The past decade has produced numerous fair learning algorithms grounded in different notions of fairness, including statistical and causality-based notions (Pedreschi et al., 2009; Hardt et al., 2016; Zhang & Bareinboim, 2018b; Wen et al., 2019; Wu et al., 2019a;c; Tian et al., 2025).

Despite this significant progress, many existing algorithms assume independent and identically distributed (IID) data (Caton & Haas, 2020; Zafar et al., 2017a; Hardt et al., 2016; Zafar et al., 2017b). In real-world scenarios, however, data instances are rarely independent and often exhibit connections, which are referred to as non-IID settings[1]. For instance, in loan-default prediction, an individual's repayment behavior may be influenced by the experiences of friends and family. To address these issues, researchers have extended fairness notions and studied fair machine learning in graphs. Group fairness notions such as demographic parity, equalized odds, and equal opportunity have been extended to graph settings (Dong et al., 2022; Fan et al., 2021; Bose & Hamilton, 2019; Zhu et al., 2024b; Luo et al., 2024; Wang et al., 2025), and new fairness notions such as degree-related fairness and node-pair distance-based fairness (Kang et al., 2020; Dong et al., 2021) have also been proposed.

However, a significant gap in the current literature on fair graph learning is the lack of principled exploration of causality-based methods. Causal fairness plays a vital role in the fair machine learning field by modeling unfairness as the causal effect of the sensitive feature on the model prediction rather than relying solely on correlation. While causality-based fairness has been extensively studied in IID settings, with notions such as direct and indirect discrimination, counterfactual fairness, and path-specific counterfactual fairness being well-established (Zhang & Bareinboim, 2018a; Chiappa, 2019; Malinsky et al., 2019; Wu et al., 2019a;c), its application in non-IID graph settings remains largely underexplored. Recent studies have highlighted that directly applying conventional fairness notions to non-IID settings without accounting for dependencies among individuals can yield biased outcomes (Zhang et al., 2022; Zhang, 2023). Although causal inference for non-IID settings has been studied in the context of interference (Hudgens & Halloran, 2008; Tchetgen & VanderWeele, 2012), integrating these methods into machine learning workflows for measuring causal unfairness in non-IID graph data remains challenging, due to computational and estimation barriers, such as the consistent interference assumption (Arbour et al., 2016a; Lee & Honavar, 2016; Arbour et al., 2016b). Preliminary work has formulated causal fairness in non-IID settings (Agarwal et al., 2022; 2021; Ma et al., 2022; Yang et al., 2024), but these studies do not provide a rigorous foundation for extending traditional inference techniques, such as *do*-calculus, to graph-structured data. A comprehensive and general theoretical framework for causal inference in such settings is still lacking.

In this work, we address a fundamental challenge in extending causal inference to graph data: the heterogeneity of causal mechanisms across nodes. In graph settings, a node's outcome is typically influenced by its own attributes and by the attributes of its neighbors. The aggregation of these influences depends on the local network structure. As a result, nodes with different neighborhood structures effectively follow different causal mechanisms, which violates the invariance assumptions that underlie classical structural causal models (SCMs) and *do*-calculus. To overcome this challenge, our key insight is that a structural representation can restore invariance even when node-level mechanisms vary with local network structure. Building on the Network Structural Causal Model (NSCM), we use the Weisfeiler-Lehman (WL) graph isomorphism framework to encode local structural information through node colors. We then introduce two general conditions, Decomposability and Graph Independence, under which the networked causal process can be reduced to an equivalent causal model with a shared mechanism across nodes. Under these conditions, interventional distributions in non-IID graph data can be expressed using standard *do*-calculus. In particular, we derive a representation of the interventional distribution that separates the effect of the sensitive attribute from network structural variability. This formulation shows that observational and interventional distributions share a common latent functional component, which can be estimated from data and reused under interventions.

Motivated by this analysis, we propose a deep learning framework, the Message Passing Variational Autoencoder for Causal Inference (MPVA), which is designed to approximate this shared component. MPVA combines message passing neural networks (MPNNs) to capture neighborhood-level effects with conditional variational autoencoders (cVAEs) to model conditional distributions. Rather than directly enforcing fairness constraints, our approach estimates interventional distributions and incorporates them into a causal fairness regularization framework for node classification.

---

[1]In this paper, the terms 'non-IID settings', 'graph settings', and 'network settings' are used interchangeably and refer to situations where data instances are interconnected.

We evaluate our approach on both semi-synthetic and real-world datasets. The results demonstrate that MPVA more accurately estimates interventional quantities in graph settings and achieves improved fairness compared to existing methods that do not account for mechanism heterogeneity.

## 2 Related Work

**Graph-based Causal Inference**. Recent work has investigated and relaxed the IID assumption in causal inference. Several studies extend graphical causal modeling frameworks. Ogburn & VanderWeele (2014) extended DAGs to represent interference relationships among individuals. Sherman & Shpitser (2018) modeled interference with chain graphs that capture unknown interactions between individuals. Bhattacharya et al. (2019) proposed an interventional method for estimating causal effects under data dependence when the structure is known. Beyond graphical modeling, researchers have defined effects that capture relationships among variables and data points. Shpitser & Pearl (2008) defined individual and group average direct and indirect effects, also known as spillover effects, under interference. Ogburn & VanderWeele (2014) developed direct interference, interference by contagion and infectiousness, and allocational interference. Recent years have also seen growing interest in causal interference effect estimation. Hudgens & Halloran (2008) and VanderWeele & Tchetgen Tchetgen (2011) developed randomized procedures for unbiased estimands. Fatemi & Zheleva (2020) proposed an experimental design approach to minimize interference bias and selection bias during estimation. Tchetgen Tchetgen et al. (2021) proposed a general g-computation method for causal interference.

**Fairness on Graphs**. Algorithmic bias in machine learning has received substantial attention, with many methods and empirical studies (Binns, 2020; Jiang et al., 2023; Verma & Rubin, 2018). Existing fairness notions can be grouped into two broad categories. The first category is *statistical parity*, which requires protected and non-protected groups to receive favorable decisions at similar rates. The quantitative metrics derived from *statistical parity* include *risk difference*, *risk ratio*, *relative change*, and *odds ratio* (Wu et al., 2019b; Hardt et al., 2016; Pedreschi et al., 2012). Recent works (Agarwal et al., 2021; Bose & Hamilton, 2019; Buyl & Bie, 2020; Dai & Wang, 2021; Dong et al., 2021; Kang et al., 2020) mitigate bias in node representation learning. Many of these methods (Beutel et al., 2017; Zhang et al., 2018) focus on adversarial learning to prevent learned representations from reliably predicting the associated sensitive attribute. These works reduce statistical dependence between the sensitive attribute and prediction in the learned representation, but they do not address bias induced by features or graph structure through causal effects. The second category is counterfactual fairness, which is developed mainly under structural causal models (SCMs) (Pearl, 2009). Several works (Ma et al., 2022; Agarwal et al., 2021; Yang et al., 2024) extend counterfactual fairness to graphs. However, most of these works ignore potential biases introduced by the sensitive attributes of neighboring nodes and by the causal effects of sensitive attributes on other nodes.

## 3 Preliminary

**Structural Causal Model (SCM)**. Structural causal models (Pearl, 2009) provide a mathematical framework to understand causal relationships within a system. SCMs define the causal dynamics of a system through a collection of structural equations. Each variable $X$ in the system is associated with a function $f_X$, such that $x = f_X(\mathsf{pa}_X, u_X)$. Here, $\mathsf{pa}_X$ denotes the values of endogenous variables that directly influence $X$, and $u_X$ represents the values of exogenous variables affecting $X$. Each SCM is associated with a causal diagram whose nodes represent variables and whose directed edges represent direct causal relations. We assume a Markovian causal model in this paper, i.e., all exogenous variables are mutually independent.

**Causality-based Fairness Notions**. Defining causality-based fairness notions is facilitated by the *do*-operator (Pearl, 2009), which simulates physical interventions that force variables to take certain values. Formally, the intervention that sets the value of $S$, a sensitive demographic characteristic (i.e., race or gender), to $s$ is denoted by $do(S = s)$. The distribution of a variable $X$ after the intervention setting $S$ to $s$ is called the interventional distribution, denoted as $P(x|do(s)) := P(x|do(S = s))$. Causality-based fairness notions are usually defined by disparities in interventional distributions across demographic groups, such as the total effect (Zhang & Bareinboim, 2018a), direct discrimination (Zhang et al., 2017), indirect discrimination (Zhang

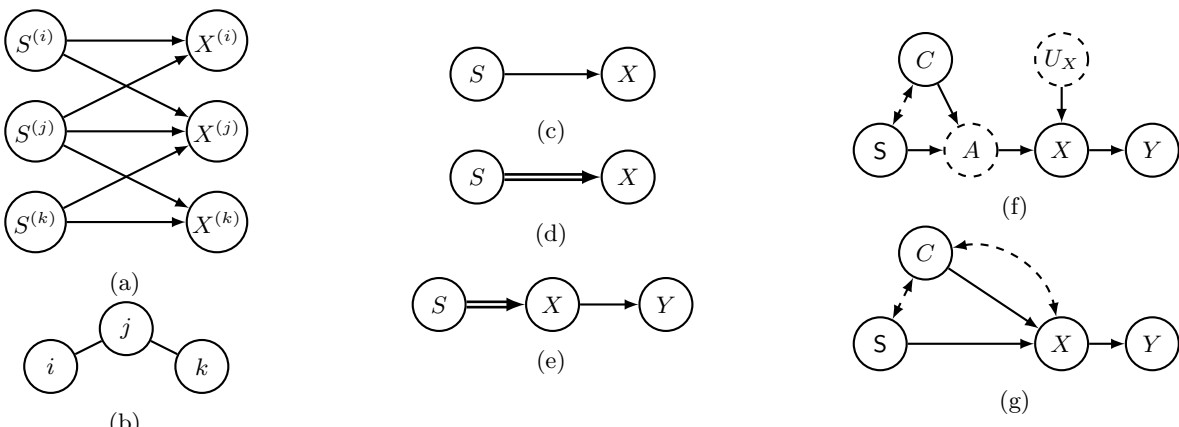

Figure 1: Graphs and diagrams. (a) The interference graph. (b) The network $\mathcal{G}$. (c) The causal diagram $\mathcal{C}$. (d) The networked causal diagram $\mathcal{N}$. (e) The networked causal diagram for node classification. (f) The causal diagram that is equivalent to the networked causal diagram in Figure 1e. (g) The causal graph that violates Graph Independence (Condition 2).

et al., 2017), and counterfactual fairness (Kusner et al., 2017). In this paper, we consider the total effect of $S$ on $Y$, defined as $\mathbb{E}[Y|do(s^+)] - \mathbb{E}[Y|do(s^-)]$, where $s^+$ and $s^-$ represent the favorable and unfavorable demographic groups.

## 4 Problem Formulation

### 4.1 Network Structural Causal Model (NSCM)

Traditional SCMs assume that data instances are IID. To handle non-IID data, recent studies have extended SCMs to capture interference between individuals (e.g., (Ogburn et al., 2022)). These extensions usually use interference graphs, as illustrated in Figure 1a, to describe both causal relationships between features and interference relationships between individuals. Motivated by this line of work, we adopt a more general framework that extends traditional SCMs, which we refer to as the Network Structural Causal Model (NSCM). In an NSCM, we combine the network, which describes potential interference, with the graph, which describes causal relationships between features. To distinguish these relationships, an NSCM explicitly considers two components: 1) a network ($\mathcal{G}$), where each node represents an individual or instance and each edge represents potential interference between connected individuals; and 2) a causal diagram ($\mathcal{C}$), where each node represents a feature and each edge represents a parent-child relationship between features determined by the structural equations. It is formally defined as follows.

**Definition 1** (Network Structural Causal Model (NSCM)). An NSCM $\mathcal{M}$ is a quadruple $\mathcal{M} = \langle \mathcal{G}, \mathbf{U}, \mathbf{V}, \mathbf{F} \rangle$ where

1. $\mathcal{G}$ is a network that consists of a set of connected nodes.
2. $\mathbf{U}$ is a set of exogenous variables. For every node $i$ in the graph, $\mathbf{u}^{(i)} \in \mathbf{U}$ is an instantiation of the exogenous variables for node $i$.
3. $\mathbf{V}$ is a set of endogenous variables. For every node $i$ in the graph, $\mathbf{v}^{(i)} \in \mathbf{V}$ is an instantiation of the endogenous variables for node $i$.
4. $\mathbf{F}$ is a set of structural equations. For each variable $X \in \mathbf{V}$ and node $i$, an equation $x^{(i)} = f(\mathsf{pa}_X^{(i)}, \{\mathsf{pa}_X^{(j)} : j \in \mathsf{ne}^{1:k}(i)\}, u_X^{(i)})$ determines the value of $x^{(i)}$, where $\mathsf{ne}^{1:k}(i)$ denotes the neighborhood of $i$ within $k$ hops.

An illustrative example of an NSCM with two variables $S, X$ is shown in Figures 1a, 1b, and 1c, which show the interference graph, the network $\mathcal{G}$, and the causal diagram $\mathcal{C}$. In this example, the structural equation of

this NSCM is

$$x^{(i)} = f(s^{(i)}, \{s^{(j)} : j \in \mathsf{ne}^{1:k}(i)\}, u_X^{(i)}) \tag{1}$$

To further simplify representation, we introduce a hybrid graph–the networked causal diagram $\mathcal{N}$–that integrates network information with the causal diagram while omitting detailed interference structure, as shown in Figure 1d. In this networked causal diagram, the solid line arrow represents the case where the causal effect is transmitted only from a variable of an individual to another variable of the same individual as seen in the traditional SCM (not shown in this example), while the double line arrow represents the existence of interference between different individuals in $\mathcal{G}$ when the causal effect is transmitted from one variable to another. For $k = 1$, the interference solely occurs between nodes that are immediate neighbors, while for $k > 1$, the interference can propagate through multiple hops in a message-passing fashion.

Given the NSCM formulation, the causal inference task is to infer the interventional distribution $P(x|do(s)) := P(x|do(S = s))$ from observational graph data.

### 4.2 Fair Node Classification

We apply our causal inference techniques to fair node classification as the downstream task. Denote the sensitive feature by $S$, the non-sensitive feature by $X$, and the decision by $Y$. We consider a general networked causal diagram as shown in Figure 1e, where, for simplicity, we posit that interference only exists from $S$ to $X$. However, our methods can be used to handle interference between any pair of variables. Suppose that we are given a dataset $\mathcal{D} = \{s^{(i)}, x^{(i)}, y^{(i)}\}_{i=1}^K$ and a network $\mathcal{N}$ that connects individuals in $\mathcal{D}$ to reflect interference. The goal is to build a classifier $h : X \mapsto Y$ that predicts the label. We say that the classifier is causally fair if $\mathbb{E}[h(x)|do(s^+)] = \mathbb{E}[h(x)|do(s^-)]$, where $do(\cdot)$ represents the intervention performed under the networked causal diagram.

## 5 Method

### 5.1 Causal Inference on Network Data

We use the networked causal diagram in Figure 1e and the task of inferring the causal effect from $S$ to $X$, i.e., computing $P(x|do(s))$, as a running example. The *do*-calculus is an axiomatic system widely used to solve causal inference problems (Pearl, 2009, Ch. 3.4). However, as shown in recent studies (Zhang et al., 2022; Zhang, 2023), directly applying *do*-calculus in non-IID settings can lead to biased results. The fundamental challenge is that, in networked data, the structural equation associated with each node depends on its local neighborhood. Unlike standard structural equations in classical SCMs, Equation 1 incorporates neighborhood information, so nodes with different local network structures follow different effective causal mechanisms. As a result, the invariance assumptions required for the validity of *do*-calculus no longer hold. This section extends the intervention *do*-operator and symbolic notation to NSCMs by identifying conditions under which a shared causal mechanism can be recovered despite neighborhood-dependent interactions.

Let $\mathsf{s}^{(i)} := \{s^{(i)}, \{s^{(j)} : j \in \mathsf{ne}^{1:k}(i)\}\}$ denote the multiset of sensitive attributes associated with node $i$ and its $k$-hop neighbors. This multiset captures all neighborhood information of $S$ that influences node $i$, including node $i$ itself. We define the intervention $do(\mathsf{s}^{(i)} = s)$ as setting the sensitive attribute of every node in $\mathsf{s}^{(i)}$ to $s$. Let $do(\mathsf{s} = s)$ denote the global intervention that applies $do(\mathsf{s}^{(i)} = s)$ simultaneously to all nodes $i$. Under this global intervention, the interventional distribution $P(x|do(\mathsf{s}) = s)$ coincides with $P(x|do(s))$ in the graph setting, which is the target quantity. The following discussion focuses on computing $P(x|do(\mathsf{s}) = s)$.

The key challenge in computing $P(x \mid do(\mathsf{s} = s))$ arises from the heterogeneity of node-level causal mechanisms induced by varying local network structures. To address this, we aim to decouple the causal effect of $S$ from the structural variability introduced by neighborhood interactions. To this end, we leverage the notion of node colors from the Weisfeiler-Lehman (WL) graph isomorphism test to encode local structural information. The node color serves as a representation of the local computation tree of each node. In particular, two nodes are assigned the same color if and only if they have identical computation trees under the WL procedure. Let $c^{(i)}$ denote the color of node $i$ obtained from the WL algorithm with identical initial node colors. The following result follows from (Jegelka, 2022):

**Lemma a** ((Jegelka, 2022)). *For two different nodes $i, j$, $c^{(i)} = c^{(j)}$ if and only if nodes $i$ and $j$ have identical computation trees in the WL graph isomorphism test.*

Based on this representation, we introduce conditions under which the heterogeneous node-level mechanisms can be reduced to a shared functional form.

**Condition 1** (Decomposability). The structural equation in the NSCM can be decomposed into a message-passing mechanism that aggregates neighborhood information and an internal mechanism that governs the node-level causal effect.

Under this condition, the structural equation $x^{(i)} = f(s^{(i)}, \{s^{(j)} : j \in \mathsf{ne}^{1:k}(i)\}, u_X^{(i)})$ can be decomposed into two equations:

$$a^{(i)} = f^{MP}(\mathsf{s}^{(i)}, c^{(i)}), \qquad x^{(i)} = f^{INT}(a^{(i)}, u_X^{(i)})$$

where $f^{MP}$ represents the message-passing mechanism, $a^{(i)}$ is an intermediate variable summarizing neighborhood effects, and $f^{INT}$ represents the internal node-level causal mechanism.

Combining Lemma a and Condition 1, we obtain the following result, which characterizes when the aggregated neighborhood effect can be represented by a shared mapping.

**Proposition 2.** *For any two nodes $i, j$, if $\{\mathsf{s}^{(i)}, c^{(i)}\} = \{\mathsf{s}^{(j)}, c^{(j)}\}$, then we have $a^{(i)} = a^{(j)}$.*

*Proof.* By Lemma a, nodes $i$ and $j$ have identical computation trees. Under Condition 1, the message-passing function $f^{MP}$ operates over these computation trees. Therefore, identical inputs and identical structural contexts imply identical outputs, i.e., $a^{(i)} = a^{(j)}$. □

**Condition 2** (Graph Independence). The exogenous variable $U_X$ is independent of node color $C$, i.e., $U_X \perp C$.

The Graph Independence condition ensures that the exogenous variation affecting $X$ does not depend on the network structure. Consequently, the intermediate variable $A$ provides a sufficient summary of all neighborhood-dependent effects relevant to $X$ under intervention.

Building on these results, we now show that, under Conditions 1 and 2, the NSCM can be reduced to an equivalent causal model that admits standard causal inference via *do*-calculus. Under Condition 1, the neighborhood-dependent structural equation can be rewritten by dropping the node index as

$$a = g(\mathsf{s}, c), \qquad x = f^{INT}(a, u_X),$$

where $c$ denotes the node color encoding local network structure, and $g$ is a deterministic mapping induced by the message-passing mechanism $f^{MP}$. This reduction transforms the original neighborhood-dependent mechanism into a two-stage process where $(\mathsf{S}, C)$ captures all structural variability from the network through $A$, and the downstream mechanism $f^{INT}$ is shared across nodes. Thus, Condition 1 recovers a shared functional form for the causal mechanism of $X$.

Given this representation, the resulting causal model consists of variables $(\mathsf{S}, C, A, X)$ with structural equations

$$c \sim P(C), \qquad \mathsf{s} \sim P(\mathsf{S} \mid C), \qquad a = g(\mathsf{s}, c), \qquad x = f^{INT}(a, u_X),$$

where the dependence between $\mathsf{S}$ and $C$ allows potential confounding induced by the network structure. These structural equations correspond to the reduced causal diagram in Figure 1f, which is a standard causal diagram without explicit interference between instances.

Condition 2 plays a critical role in identifiability because the assumption $U_X \perp C$ guarantees that all graph-dependent effects on $X$ are mediated through $A$. This assumption prevents additional confounding paths from $C$ to $X$ through the exogenous variable. In the reduced causal diagram, the only backdoor path from $\mathsf{S}$ to $X$ is $\mathsf{S} \leftrightarrow C \to A \to X$, which arises from the dependence between $\mathsf{S}$ and the structural variable $C$. Conditioning on $C$ blocks this backdoor path, so $C$ is a valid adjustment set.

We then examine the case where Condition 2 is violated. In this setting, the dependence between $U_X$ and $C$ induces latent confounding between $C$ and $X$, which can be represented as a bidirected edge $C \leftrightarrow X$ in the

causal diagram. To analyze identifiability, we consider the latent projection of the model onto the observed variables $\{S, C, X\}$ by marginalizing out the latent variables $A$ and $U_X$ (see Figure 1g). In the resulting graph, the directed edges reduce to $S \to X$ and $C \to X$, while latent confounding induces bidirected edges $S \leftrightarrow C$ and $C \leftrightarrow X$. Consequently, the variables $S$, $C$, and $X$ belong to a single c-component. Under this structure, the Hedge Criterion (Shpitser & Pearl, 2008) can be applied to assess identifiability. In particular, consider the two c-forests: (i) the larger forest $F' = \{S, C, X\}$, which is rooted at $X$ and contains the treatment variable $S$, and (ii) the smaller forest $F = \{C, X\}$, which is also rooted at $X$ but excludes $S$. Since $F \subset F'$ and both forests share the same root, the hedge criterion is satisfied. By the completeness of the hedge criterion, this structure implies that the causal effect $P(x \mid do(S) = s)$ is not identifiable from observational data in general.

As a result, we have the following proposition.

**Proposition 3.** *Under Conditions 1 and 2, the causal effect $P(x \mid do(S) = s)$ is identifiable via adjustment on $C$, provided that $C$ blocks all backdoor paths from $S$ to $X$ and $P(S = s \mid C = c) > 0$ for all $c$ with $P(C = c) > 0$.*

## 5.2 From Identification to Estimation

While Section 5.1 establishes that the causal effect $P(x \mid do(S) = s)$ is identifiable via adjustment on the structural variable $C$, directly estimating the conditional distribution $P(x \mid s, c)$ for the adjustment remains challenging in practice. In particular, the node color $C$ encodes fine-grained local graph structure and may take a large number of distinct values, which leads to high-dimensional and sparsely supported conditioning.

To address this issue, we leverage the decomposable structure implied by Condition 1. Specifically, under this condition, the effect of $(S, C)$ on $X$ is mediated through an intermediate variable $A = g(S, C)$, which provides a compact representation of neighborhood influence. This structure suggests replacing direct conditioning on $(S, C)$ with a learned representation $A$ that captures the structural information relevant to predicting $X$.

Motivated by this observation, we derive a representation of the interventional distribution that separates structural variability from the downstream causal mechanism. This formulation provides a principled route for estimating $P(x \mid do(S) = s)$ and directly informs our model architecture, as described below.

**Theorem 4.** *Suppose Conditions 1 and 2 hold. Then the interventional distribution of $X$ under $do(S) = s$ admits the following representation:*

$$P(x \mid do(S) = s) = \sum_c P(c) \sum_a P(x \mid a) P(g(S, c) = a), \tag{2}$$

*where $A = g(S, C)$ denotes the intermediate variable induced by the message-passing mechanism.*

Theorem 4 provides a representation of the interventional distribution through the intermediate variable $A = g(S, C)$. To motivate estimation, we next relate this expression to the observational distribution.

By marginalizing over $S$, $C$, and $A$, the observational distribution of $X$ can be written as

$$P(x) = \sum_{S,c,a} P(S, c) P(g(S, c) = a) P(x \mid a) = \sum_{S,c} P(S, c) \sum_a P(x \mid a) P(g(S, c) = a). \tag{3}$$

Comparing Equation 2 from Theorem 4 with Equation 3, we observe that both the interventional and observational distributions share the same inner quantity $Q(x; S, c) := \sum_a P(x \mid a) P(g(S, c) = a)$. The key implication is that the observational and interventional distributions differ only in how the shared component $Q(x; S, c)$ is averaged. In the observational setting, $Q(x; S, c)$ is averaged with respect to the joint distribution $P(S, c)$. Under intervention, $S$ is fixed, and the averaging is performed only over the structural variable $C$ according to $P(c)$.

This observation admits a natural Monte Carlo interpretation. Suppose a model is learned to approximate the shared component $Q(x; S, c)$. The observational distribution $P(x)$ can then be estimated by sampling pairs $(S, c) \sim P(S, c)$ and averaging the resulting values of $Q(x; S, c)$. The interventional distribution $P(x \mid do(S) = s)$

can be estimated by fixing $\mathsf{s}$, sampling $c \sim P(c)$, and averaging the same quantity $Q(x; \mathsf{s}, c)$. Once the shared component is learned, intervention amounts to reweighting or resampling rather than learning a new model.

Based on this observation, we state the following estimation principle.

**Corollary 5.** *Suppose a model $Q_\theta(x; \mathsf{s}, c)$ is learned from observational data to approximate the shared component $\sum_a P(x \mid a) P(g(\mathsf{s}, c) = a)$ in Equation 3. Assume further that the learned mapping remains valid under interventions on $\mathsf{S}$ in the sense that the intervention changes only the weighting over $(\mathsf{s}, c)$, while preserving the structural mechanism encoded by $Q_\theta$. Then the interventional distribution can be approximated by*

$$P(x \mid do(\mathsf{s}) = s) \approx \sum_c P(c) \, Q_\theta(x; \mathsf{s}, c) = \mathbb{E}_{c \sim P(c)}[Q_\theta(x; \mathsf{s}, c)]. \tag{4}$$

Corollary 5 does not require explicitly recovering the true latent values of $A$. Instead, it suggests learning a representation that captures the shared functional component relating $(\mathsf{S}, C)$ to $X$ and reusing this representation under intervention through a modified averaging scheme. This result provides a practical route for estimating interventional distributions while remaining consistent with the decomposable structure in Condition 1.

Motivated by this principle, we design the Message Passing Variational Autoencoder (MPVA) for causal inference in networks, which combines a message-passing neural network (MPNN) with a conditional variational autoencoder (cVAE), as detailed in the next subsection.

## 5.3 Message Passing Variational Autoencoder for Causal Inference (MPVA)

We develop the MPVA framework to directly implement the estimation principle established in Theorem 4. The architecture is designed to approximate the shared component $Q(x; \mathsf{s}, c)$ between observational and interventional distributions, which allows interventional distributions to be computed through reweighting. For generality, we further consider the existence of independent variables $Z$ other than $S$ that directly affect $X$, and Corollary 5 readily applies to this case. In this framework, we first use an MPNN to learn the intermediate representation $A$ under Condition 1, which captures the aggregated causal effect of $S$ on $X$ from each node's neighborhood through the message-passing mechanism $g(\mathsf{s}, c)$, denoted as $\hat{a} = \hat{g}^{MPNN}(\mathsf{s})$. Then, we use the estimated intermediate representation $\hat{A}$ along with the variables $Z$ as inputs to a multilayer perceptron (MLP) to predict the node feature $X$, denoted as $\hat{x} = MLP(\hat{a}, z)$. We then use a cVAE to learn the conditional distribution $P(x|\hat{a}, z)$, which corresponds to $P(x|a)$ in Theorem 4. The encoder $v = EN(x, \hat{a}, z)$ takes $x$ as input with $\hat{a}, z$ as conditions, and the decoder $\hat{x} = DE(v, \hat{a}, z)$ reconstructs $x$. The architecture of MPVA is illustrated in Figure 2.

In the training phase, we first train the MPNN and MLP with the dataset $\mathcal{D}$ and network $\mathcal{N}$. Note that this process does not require actual values of $A$ as supervised signals. Then, we freeze the parameters of MPNN and MLP and train the cVAE. In the inference phase, for each node $i$, we first use the MPNN to compute $\hat{a}^{(i)} = \hat{g}^{MPNN}(\mathsf{s}^{(i)})$ and use the encoder of the cVAE to compute $v^{(i)} = EN(x^{(i)}, \hat{a}^{(i)}, z^{(i)})$. Then, we perform the intervention $do(\mathsf{s}) = s$ to change the value of $S$ to $s$ for all nodes. Next, we use the MPNN to recompute $A$ under the intervention, i.e., $\tilde{a}_s = \hat{g}^{MPNN}(do(\mathsf{s}^{(i)}) = s)$, and use the cVAE to reconstruct $X$ from $\tilde{a}_s$ and $v^{(i)}$, i.e., $\tilde{x}_s^{(i)} = DE(v^{(i)}, \tilde{a}_s, z^{(i)})$. Thus, $\tilde{x}_s^{(i)}$ represents the estimated outcome for node $i$ under the population-level intervention $do(\mathsf{s}) = s$. This inference process follows an Abduction-Action-Prediction structure: it encodes the observed outcome (abduction), computes the intervention effect on the intermediate representation (action), and decodes to obtain the interventional outcome (prediction). Critically, Conditions 1 and 2 guarantee the identifiability of these interventional distributions, as established in Section 5.1.

## 5.4 Causally Fair Node Classification

Having established a method for estimating interventional distributions, we now apply it to causally fair node classification. We formulate causally fair node classification as a regularized optimization problem that explicitly minimizes the causal effect of sensitive attributes on predictions. To this end, we use the interventional distributions estimated by MPVA to define a causal fairness regularization term, which is added to the standard classification loss. Specifically, we generate interventional outcomes using the classifier

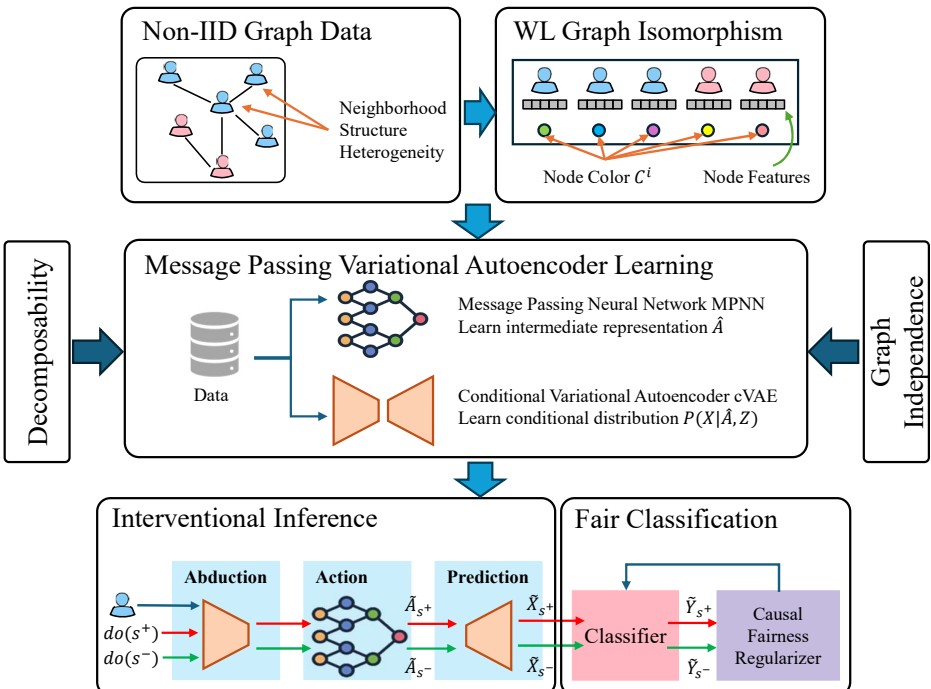

Figure 2: Overview of the MPVA framework. The MPNN component learns node-level representations by aggregating causal influences from neighboring nodes. The cVAE models the conditional distribution necessary to estimate interventional outcomes. Together, the MPNN and cVAE enable inference of the interventional distribution of $Y$ and act as a regularizer to improve causally fair classification.

$h$ and the learned MPVA, denoted by $\tilde{y}_{s+}$ and $\tilde{y}_{s-}$, where $\tilde{y}_s = h(\tilde{x})|do(s)$. For the fair classification task, we construct a regularization term that minimizes the causal discrepancy between two interventional variants:

$$\ell_f = \mathbb{E}[h(\tilde{x})|do(s^+)] - \mathbb{E}[h(\tilde{x})|do(s^-)]$$
$$= \mathbb{E}[\mathbb{1}_{\tilde{y}_{s+}=1}] - \mathbb{E}[\mathbb{1}_{\tilde{y}_{s-}=1}] = \mathbb{E}[\mathbb{1}_{\tilde{y}_{s+}=1}] + \mathbb{E}[\mathbb{1}_{\tilde{y}_{s-}=-1}] - 1, \tag{5}$$

where $\mathbb{1}$ is the indicator function. Following (Wu et al., 2019b), the indicator function can be further replaced with the differentiable surrogate function $u(\cdot)$. It is noteworthy that this differentiability allows the regularization term to be incorporated into the classic loss functions used for training the node classifier. Thus, this regularization term can be seamlessly incorporated into the classification loss: $\ell = \frac{1}{K} \sum_{i \in [K]} \ell_c(h(x^{(i)}), y^{(i)}) + \lambda \ell_f$, where $\ell_c$ is the empirical loss function and $\lambda$ is a hyperparameter that balances model performance and causal fairness.

## 6 Experiments

We evaluate the proposed method and baselines on semi-synthetic and real-world graphs. To support reproducibility, we provide implementation details and experimental settings in the appendix. The complete source code for MPVA, including the Message Passing Variational Autoencoder architecture and training procedures, is available at a public repository[2]. All hyperparameter settings, network architectures, and optimization details are specified in the appendix. All baseline implementations follow their original papers and publicly available code. We report means and standard deviations across five independent runs.

---

[2]http://tiny.cc/mpva

## 6.1 Datasets

In our experiments, we use both semi-synthetic datasets, which allow full control over the data-generation process, and real-world datasets, which evaluate external generalizability. Semi-synthetic data are commonly used in causal inference and fair machine learning because ground-truth quantities, such as causal effects and bias, cannot be directly observed in real-world settings. To create semi-synthetic datasets, we adapt the Credit Dataset (Yeh, 2016) by introducing network structures and causal relationships using the Network Structural Causal Model. This approach allows us to precisely control data generation and accurately derive ground-truth interventional distributions for arbitrary interventions on the sensitive attribute. Specifically, we generate two semi-synthetic datasets, denoted as D1 and D2, for evaluation purposes. The synthetic data generation process, which follows the Network Structural Causal Model, is fully described in the appendix. We also conduct experiments on widely used real-world datasets, namely Credit Defaulter (Yeh, 2016) and German (Hofmann, 1994). For real-world datasets (Credit and German), we provide detailed preprocessing steps and train/validation/test splits in the supplementary code repository. Since the underlying mechanisms of the real-world datasets are unknown, we evaluate our framework's ability to estimate non-IID causal interventions and compare it against baseline methods, including IID-based causal fairness approaches. We use the learned MPVA model to measure the interventional quantities and evaluate the performance in terms of non-IID causal fairness. The detailed statistics of these datasets are included in the appendix, including the number of nodes, the number of edges, and the feature dimension.

## 6.2 Experiment Settings

**Fairness Metrics:** We evaluate the performance of the proposed framework in terms of prediction accuracy and fairness. For non-causal fairness notions, we use demographic parity, a widely used fairness notion, to evaluate group-level fairness. Demographic parity requires classifier decisions to be independent of the sensitive attribute. It is usually quantified by *risk difference* (**RD**), i.e., the difference in positive prediction rates between the favorable and unfavorable groups. It can be expressed as $\left| \mathbb{E}_{X|S=s^+}[\hat{Y}] - \mathbb{E}_{X|S=s^-}[\hat{Y}] \right|$. For causal fairness notions, we consider both IID and non-IID, i.e., graph-based, causal fairness notions. We denote the IID causal fairness as **CF** whose calculation and estimation approaches are described in the appendix. We denote the graph-based causal fairness notion by **gCF**, as described in Equation 5.

**Mitigation Baselines:** We compare the proposed framework **MPVA** with several state-of-the-art non-IID bias mitigation methods and the conventional IID constraint-based methods. **GCN-RD** and **GCN-IID** are conventional Graph Convolutional Networks (GCNs) with risk-difference and IID causal fairness constraints, respectively. The constraint formulation is described in the appendix. **FairGNN** (Dai & Wang, 2021) uses a covariance-based adversarial discriminator to predict the sensitive attribute from the conventional GNN node classifier. **GEAR** (Ma et al., 2022) uses a variational autoencoder to synthesize counterfactual samples and achieve counterfactual fairness for node classification. **NIFTY** (Agarwal et al., 2021) enhances fairness and stability in GNNs by introducing a novel objective function and layer-wise weight normalization based on the Lipschitz constant. **FairINV** (Zhu et al., 2024a) trains fair GNNs by eliminating spurious correlations between labels and sensitive attributes within a single training session. Implementation details appear in Section C.2 in the appendix.

Table 1: Fairness measurement of conventional GCN using various metrics on semi-synthetic datasets.

| Data | Acc | RD | CF | gCF | True gCF |
|------|-----|-----|-----|------|----------|
| Semi-synthetic D1 | $0.9674_{\pm 0.0017}$ | $0.0580_{\pm 0.0015}$ | $0.0319_{\pm 0.0001}$ | $0.1792_{\pm 0.0344}$ | $0.1960_{\pm 0.0338}$ |
| Semi-synthetic D2 | $0.9715_{\pm 0.0011}$ | $0.0832_{\pm 0.0013}$ | $0.0461_{\pm 0.0003}$ | $0.6721_{\pm 0.0132}$ | $0.6884_{\pm 0.0084}$ |

## 6.3 Results on Semi-synthetic Data

We first generate two network structures with different generating parameters for the semi-synthetic datasets. To show the biases in the generated graphs under the proposed metrics and evaluate the effectiveness of MPVA for estimating gCF, we train the classic GCN models without any bias mitigation considerations.

Table 2: Evaluation of mitigation methods on semi-synthetic datasets.

| Data | Metric | Own Metric | Acc | gCF | True gCF |
|---|---|---|---|---|---|
| Semi-synthetic D1 | MPVA | - | $0.9259_{\pm 0.0051}$ | $\mathbf{0.0023_{\pm 0.0012}}$ | $\mathbf{0.0010_{\pm 0.0006}}$ |
| | GCN-RD | $0.0074_{\pm 0.0074}$ | $0.8743_{\pm 0.0179}$ | $0.4150_{\pm 0.0961}$ | $0.4219_{\pm 0.0969}$ |
| | GCN-IID | $0.0031_{\pm 0.0015}$ | $0.9024_{\pm 0.0071}$ | $0.1787_{\pm 0.0437}$ | $0.1896_{\pm 0.0456}$ |
| Semi-synthetic D2 | MPVA | - | $0.9490_{\pm 0.0009}$ | $\mathbf{0.0019_{\pm 0.0018}}$ | $\mathbf{0.0073_{\pm 0.0048}}$ |
| | GCN-RD | $0.0091_{\pm 0.0055}$ | $0.8828_{\pm 0.0026}$ | $0.1634_{\pm 0.0744}$ | $0.1868_{\pm 0.0900}$ |
| | GCN-IID | $0.0069_{\pm 0.0010}$ | $0.9094_{\pm 0.0021}$ | $0.6818_{\pm 0.0310}$ | $0.6977_{\pm 0.0292}$ |

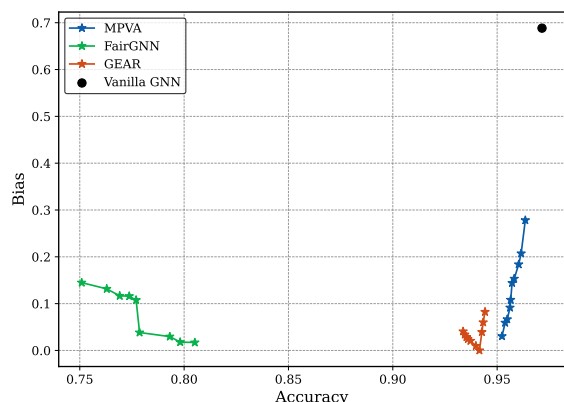

Figure 3: Comparison of measured bias and true gCF bias on D2 with various mitigation methods.

For the conventional GCN node classifiers, we measure bias and report the results in Table 1. The table presents empirical node prediction accuracy, estimated RD, CF, and gCF (highlighted in blue), as well as ground-truth gCF (highlighted in green), which is computed directly by performing interventions on the true causal model. The node prediction accuracy is high, which indicates that the models are well trained and make accurate predictions. The estimated gCF (blue) closely matches ground-truth gCF (green), showing that our method accurately estimates causal fairness in graph data. We also observe that RD and CF differ substantially from gCF, which shows that RD and CF cannot be used as proxies for gCF.

Next, we build fair node classification models on the generated graph data using the proposed method and baselines. The performance of classification prediction and fairness is shown in Table 2. For a fair comparison, each model is trained with its own fairness metric (i.e., GCN-RD uses RD, and GCN-IID uses CF). As shown in the **Own Metric** column, all models achieve low bias under their own metrics. For MPVA, the own metric is shown in the gCF column. We then report the accuracy, estimated gCF, and ground-truth gCF of all methods. Although the baselines, including GCN-RD and GCN-IID, are fair under their own metrics, they exhibit substantial bias from the *graph-based* causal fairness, i.e., gCF, perspective. In addition, the baseline methods neglect potential graph-structure effects while addressing bias, which compromises accuracy. In contrast, MPVA achieves the best performance in terms of both accuracy and graph-based causal fairness gCF. We further compare our proposed MPVA with

Figure 4: Trade-off between fairness and accuracy for MPVA, FairGNN, and GEAR on the semi-synthetic dataset D2.

the state-of-the-art graph-based bias mitigation algorithms, FairGNN and GEAR. FairGNN aims to mitigate statistical bias in graph data, while GEAR aims to alleviate counterfactual bias in graph data. For a comprehensive comparison, we evaluate the trade-off between model bias and performance for MPVA, FairGNN, and GEAR. As shown in Figure 3, we tune each model multiple times to obtain different bias-accuracy trade-offs. Each subplot reports the corresponding fairness or bias measure used by the models and the true gCF derived from the data-generation process at different accuracy levels. For MPVA, the estimated fairness aligns with

true causal fairness gCF at every accuracy level because of our graph causal inference technique. However, for FairGNN, and GEAR, the measured fairness is significantly different from the true fairness, implying that they cannot guarantee obtaining a fair model by fine-tuning models to balance the bias-accuracy trade-off. In addition, we compare the trade-off between fairness and accuracy for each method on the dataset D2, as shown in Figure 4. The figure demonstrates that MPVA achieves the most favorable trade-off, attaining high accuracy while maintaining low gCF bias. In contrast, FairGNN and GEAR struggle to simultaneously achieve both low gCF bias and accuracy, further validating the effectiveness of our causal framework in balancing both objectives.

## 6.4 Sensitivity Analysis

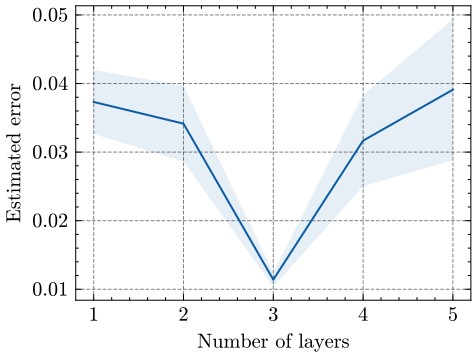

Figure 5: The impact of the number of MPNN layers on estimation error.

Table 3: gCF difference under different dependency strengths.

| Violation Strength | gCF Difference |
|:---:|:---:|
| 0.0 | $0.0075_{\pm 0.0007}$ |
| 0.5 | $0.0592_{\pm 0.0012}$ |
| 1.0 | $0.1142_{\pm 0.0016}$ |

### 6.4.1 Multi-hop Causal Effect Analysis

We further demonstrate that the proposed MPVA framework is capable of capturing multi-hop causal effects in graph data. We generate the influence of neighbors' sensitive attributes on each node within a three-hop range, based on the dataset D2. As shown in Figure 5, when the MPNN module has the same number of layers as the neighborhood hops of the generating model (which is 3 in Figure 5), it achieves the best performance in estimating the interventional distribution. This occurs because using too few layers leads to underfitting, while using too many layers results in overfitting. We also observe that the variance of the results is minimal when the number of layers matches the neighborhood hops. This observation guides the selection of an appropriate number of layers in practice.

### 6.4.2 Violation of Proposed Conditions

Additionally, we simulate a scenario where the proposed conditions are violated using the synthetic dataset D2. To facilitate the analysis, we introduce a dependency between $U_X$ and $A$ by incorporating a weighted multiplicative interaction (e.g., $\rho \cdot A \cdot U_X$) during the data generation process, with coefficient $\rho$ controlling the violation or dependency strength. We then measure the absolute difference between our estimated gCF and the True gCF. As shown in Table 3, the difference increases with violation strength, which indicates that deviations from the proposed conditions affect estimation accuracy.

## 6.5 Results on Real Data

We further conduct extensive experiments on real-world data. We first train a standard Graph Convolutional Network (GCN) without any bias mitigation method, then run fairness-aware methods over five independent trials. The results are shown in Table 4. In the original GCN, the gap between gCF and CF is large, which implies that the IID causal metric does not accurately measure non-IID causal fairness in graphs. On both datasets, MPVA **outperforms** all baselines in terms of gCF with a mild accuracy decrease compared to the classic GCN model. Although other methods achieve fairness under their own metrics, they fail to meet the gCF requirements. FairGNN neglects causality-based bias, which compromises accuracy when pursuing

Table 4: Results of various methods on real-world datasets.

| Dataset | Method | Acc | RD | CF | gCF |
|---|---|---|---|---|---|
| Credit | GCN | $0.8192_{\pm 0.0005}$ | $0.0195_{\pm 0.0014}$ | $0.0049_{\pm 0.0001}$ | $0.0705_{\pm 0.0123}$ |
| | GCN-RD | $0.7988_{\pm 0.0062}$ | $0.0057_{\pm 0.0044}$ | $0.0055_{\pm 0.0001}$ | $0.0540_{\pm 0.0145}$ |
| | FairGNN | $0.7930_{\pm 0.0086}$ | $0.0047_{\pm 0.0012}$ | $0.0043_{\pm 0.0008}$ | $0.0404_{\pm 0.0251}$ |
| | GCN-IID | $0.8065_{\pm 0.0008}$ | $0.0100_{\pm 0.0023}$ | $0.0010_{\pm 0.0003}$ | $0.1360_{\pm 0.0833}$ |
| | NIFTY | $0.7933_{\pm 0.0146}$ | $0.0543_{\pm 0.0068}$ | $0.0079_{\pm 0.0006}$ | $0.0981_{\pm 0.0165}$ |
| | GEAR | $\mathbf{0.8075_{\pm 0.0005}}$ | $0.0260_{\pm 0.0108}$ | $0.0055_{\pm 0.0003}$ | $0.0278_{\pm 0.0111}$ |
| | FairINV | $0.7720_{\pm 0.0205}$ | $0.0111_{\pm 0.0139}$ | $0.0087_{\pm 0.0003}$ | $0.0295_{\pm 0.0226}$ |
| | MPVA | $0.8054_{\pm 0.0033}$ | $0.0142_{\pm 0.0046}$ | $0.0075_{\pm 0.0007}$ | $\mathbf{0.0036_{\pm 0.0033}}$ |
| German | GCN | $0.9758_{\pm 0.0027}$ | $0.0771_{\pm 0.0083}$ | $0.0704_{\pm 0.0004}$ | $0.6080_{\pm 0.2596}$ |
| | GCN-RD | $0.9608_{\pm 0.0054}$ | $0.0046_{\pm 0.0023}$ | $0.0354_{\pm 0.0005}$ | $0.2657_{\pm 0.0600}$ |
| | FairGNN | $0.8183_{\pm 0.0162}$ | $0.0051_{\pm 0.0038}$ | $0.0536_{\pm 0.0012}$ | $0.8263_{\pm 0.0526}$ |
| | GCN-IID | $0.7643_{\pm 0.0118}$ | $0.1719_{\pm 0.0133}$ | $0.0047_{\pm 0.0011}$ | $0.2994_{\pm 0.0317}$ |
| | NIFTY | $0.8113_{\pm 0.0224}$ | $0.0975_{\pm 0.0347}$ | $0.0566_{\pm 0.0011}$ | $0.9987_{\pm 0.0019}$ |
| | GEAR | $\mathbf{0.9717_{\pm 0.0187}}$ | $0.0689_{\pm 0.0275}$ | $0.0359_{\pm 0.0040}$ | $0.3667_{\pm 0.3769}$ |
| | FairINV | $0.8200_{\pm 0.0433}$ | $0.0428_{\pm 0.0508}$ | $0.0607_{\pm 0.0063}$ | $0.5844_{\pm 0.3669}$ |
| | MPVA | $0.9283_{\pm 0.0353}$ | $0.1136_{\pm 0.0332}$ | $0.0667_{\pm 0.0014}$ | $\mathbf{0.0030_{\pm 0.0032}}$ |

fairness. The baseline GEAR achieves good accuracy but fails to eliminate bias effectively. For example, on the German dataset, GEAR does not remove the substantial gCF bias. These results show that existing fairness methods cannot guarantee gCF fairness. Overall, the real-world results are consistent with the semi-synthetic results and demonstrate the superiority of the proposed method.

# 7 Conclusions

This paper addressed a fundamental challenge in extending causal fairness to graph data: the heterogeneity of causal mechanisms across nodes due to varying neighborhood structures, which violates the invariance assumptions underlying classical structural causal models. We focused on graph settings where data instances are interconnected and proposed a principled solution based on the Network Structural Causal Model (NSCM) framework. Our key insight is that, although node-level mechanisms vary with local network structure, it is feasible to construct a structural representation that restores invariance. We introduced two conditions, Decomposability and Graph Independence, that formalize when interventional distributions can be identified using *do*-calculus in non-IID settings by separating the effect of sensitive attributes from network structural variability. Building on this theoretical foundation, we developed the Message Passing Variational Autoencoder for Causal Inference (MPVA), which estimates the shared functional component between observational and interventional distributions. We further integrated MPVA with a causal fairness regularization framework that explicitly minimizes the causal effect of sensitive attributes on predictions through interventional distribution estimation. This integration yields causally fair node classification in non-IID graph settings. Empirical evaluations on semi-synthetic and real datasets show that our approach surpasses baseline methods by more accurately approximating interventional distributions and reducing bias. Sensitivity analysis further shows that MPVA captures multi-hop causal effects and maintains performance under varying conditions.

# 8 Limitations and Future Work

While our proposed method has shown promising results, several limitations warrant discussion, along with potential directions for future work. Our method assumes that the causal graph structure is known a priori. In practice, the true causal graph is often unknown and must be inferred from data. Future work could integrate causal discovery algorithms to automatically learn the underlying causal structure, or develop robust methods that are less sensitive to misspecification of the causal graph. In addition, the current framework is specifically designed for graph-structured data where dependencies between instances are explicitly modeled through edges. Extending this approach to other settings, such as tabular data with latent dependencies or temporal data with sequential dependencies, represents an important direction. This could involve developing analogous

principles for decomposability and independence in these alternative settings. Our current formulation primarily focuses on scenarios with a single sensitive attribute. Real-world fairness problems often involve multiple intersecting sensitive attributes (e.g., race and gender). Extending the framework to handle multiple sensitive attributes by leveraging intersectional fairness (Foulds et al., 2020) is an important avenue for future research.

## Acknowledgments

This work was supported in part by NSF 1910284, 2142725, 2242812, 2520496, and SC EPSCoR 24-GA02.

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

## Appendix

## A  Proof of Theorem 4 in the main paper

For any two variables $X$ and $Y$ in an SCM, let $y(u)$ be the value of $Y$ for an instance whose exogenous variable is $u$, and let $y_x(u)$ be the value of $Y$ under the intervention $x(u) = x$. According to Pearl (2009), we have the following lemmas.

**Lemma f.** *Given the causal diagram in Figure 1f in the main paper, we have that $X_a \perp A$ for any $a$.*

**Lemma g.** *Given the causal diagram in Figure 1f in the main paper, for any node $u$, we have that $x_\mathsf{s}(u) = x_{\mathsf{s},c,a}(u) = x_a(u)$ if $a_\mathsf{s}(u) = a$ and $c(u) = c$.*

**Lemma h.** *Given the causal diagram in Figure 1f in the main paper, for any node $u$, we have that $a_\mathsf{s}(u) = a_{\mathsf{s},c}(u)$ if $c(u) = c$.*

**Theorem 9.** *Given the causal diagram in Figure 1f in the main paper, we have*

$$P(x|do(\mathsf{s}) = s) = \sum_c P(c) \sum_a P(x|a)P(g(\mathsf{s},c) = a).$$

*Proof.* According to the formula of the conditional probability, we directly have

$$P(x|do(\mathsf{s}) = s) = \sum_{c,a} P(x|c,a,do(\mathsf{s}) = s)P(c,a|do(\mathsf{s}) = s)$$

$$= \sum_{c,a} P(x|c,a,do(\mathsf{s}) = s)P(c|do(\mathsf{s}) = s)P(a|c,do(\mathsf{s}) = s).$$

Since $\mathsf{S}$ is not a descendant of $C$ in the causal diagram, it follows that

$$P(x|do(\mathsf{s}) = s) = \sum_{c,a} P(x|c,a,do(\mathsf{s}) = s)P(c)P(a|c,do(\mathsf{s}) = s).$$

According to Lemma h, we have $P(a|c,do(\mathsf{s}) = s) = P(a|do(c),do(\mathsf{s}) = s)$ which can be rewritten as $P(g(\mathsf{s},c) = a)$ below using the mapping $g$. By similarly applying Lemma g, we have $P(x|c,a,do(\mathsf{s}) = s) = P(x|do(c),do(a),do(\mathsf{s}) = s) = P(x|do(a))$. As a result, we have

$$P(x|do(\mathsf{s}) = s) = \sum_{c,a} P(x|do(a))P(c)P(g(\mathsf{s},c) = a).$$

Then, we rewrite the above equation as

$$P(x|do(\mathsf{s}) = s) = \sum_{c,a} \sum_{a'} P(x|a',do(a))P(a')P(c)P(g(\mathsf{s},c) = a)$$

According to Lemma f, we have $P(x|a',do(a)) = P(x|a,do(a))$, which is equal to $P(x|a)$ according to the Composition Axiom. Finally, we have that

$$P(x|do(\mathsf{s}) = s) = \sum_{c,a} P(x|a) \sum_{a'} P(a')P(c)P(g(\mathsf{s},c) = a)$$

$$= \sum_c P(c) \sum_a P(x|a)P(g(\mathsf{s},c) = a).$$

Hence, the theorem is proved. $\qquad\square$

## B    Discussion of Proposed Conditions

### B.1    Decomposability

The Decomposability condition states that when a node's outcome is influenced by both its own attributes and its neighbors' attributes, the influence can be decomposed into two steps: aggregating information from neighbors and then combining it with the node's own attributes. This condition makes causal inference tractable in graph settings with a structured model of information flow through the network. Under this condition, we can decompose the causal effect of neighbors into the message-passing mechanism, denoted by $f^{MP}(\cdot)$, and the internal causal mechanism within the node, denoted by $f^{INT}(\cdot)$.

For example, consider a social media platform that uses an algorithm to deliver purchase discounts to users. We posit that a user may choose to subscribe to the supplier (i.e., $X$), and if they decide to subscribe to the supplier, the chance of receiving the discount will increase. Due to the connections in the social network, we posit that a user's decision to subscribe is influenced by their own situation (i.e., $S$) as well as their neighbors. Decomposability posits that the user first aggregates information from all neighbors and then combines it with their own situation when making the decision. This two-step process allows us to separately model the network effects and individual effects while still capturing their joint influence on the final outcome.

### B.2    Graph Independence

The Graph Independence condition posits that the graph structure (captured by the node color $C$) is independent of the exogenous variable (denoted as $U_X$), which ensures that all relevant information from the neighborhood regarding the interventional value of $X$ can be effectively summarized in the intermediate variable $A$. Together, Proposition 2 and Condition 2 imply that we can convert the networked causal diagram in Figure 1e (from the main paper) to an equivalent causal diagram as shown in Figure 1f (from the main paper).

In the example of a social network where users are connected based on shared interests or demographics, the Graph Independence condition means that the network connections themselves (who is friends with whom) are not influenced by exogenous factors that also affect the non-sensitive attributes. This condition allows us to treat the network structure as fixed and unaffected by interventions, thereby preventing cyclic dependencies in the causal diagram (i.e., Figure 1f). In other words, when we intervene on a node's attributes, we posit that this intervention does not alter the underlying network topology.

## C    Experiments

### C.1    Datasets

For the semi-synthetic datasets, we leverage the Credit Dataset (Yeh, 2016). To obtain full control over the data generation process, we define the data generation mechanism as follows. Denoting the original features in the dataset as $O$, we first build a classifier $f_s : \mathcal{O} \to [0, 1]$ to predict the sensitive attribute $S$ from $O$ and use its outputs to specify Bernoulli draws of $s^{(i)}$ at each node $i$. We similarly build a classifier $f_y : \mathcal{X} \to \mathcal{Y}$ to model $y^{(i)}$ from $x^{(i)}$. We then randomly initialize a GNN $g(\cdot)$ to mimic the influence of neighbors' sensitive attributes on each node's non-sensitive attributes $X$. We generate our semi-synthetic dataset as follows:

$$s^{(i)} \sim \text{Bernoulli}(p)$$
$$x^{(i)} = g(\mathsf{s}^{(i)}) + O^{(i)} + \xi^{(i)}$$
$$y^{(i)} = f_y(x^{(i)})$$

where $p = f_s(O^{(i)})$ is the conditional probability of $s^{(i)} = 1$, $\xi^{(i)}$ is Gaussian noise representing the exogenous disturbance $u_X^{(i)}$, and $g(\mathsf{s}^{(i)})$ is the neighborhood shift returned by the GNN from the sensitive-attribute inputs $\mathsf{s}^{(i)}$. Crucially, the draws of $\xi^{(i)}$ and the randomly initialized GNN $g(\cdot)$ (which specifies the neighborhood influence structure) are generated independently. This independence ensures that the Graph Independence

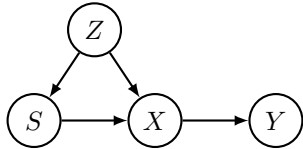

Figure 6: The networked causal diagram for node classification.

condition (Condition 2) holds. We generate ground truth interventional data under $do(\mathsf{s} = 1)$ and $do(\mathsf{s} = 0)$ to obtain positive and negative interventional distributions, respectively.

For the real-world graphs, we conduct experiments on widely used real-world datasets, namely Credit Defaulter (Yeh, 2016) and German (Hofmann, 1994). The dataset details are as follows.

**Credit Defaulter**: This dataset contains 30,000 instances representing credit card users, and graph connections/edges are formed based on the similarity of payment information (a subset of features in this dataset). The task is to predict default payment with the sensitive attribute "Sex". We treat "Education", "Marriage", and "Age" as $Z$ (variables other than $S$ that directly affect $X$), and the remaining features as $X$.

**German**: This dataset contains 1,000 instances representing clients of a German bank, and graph connections/edges are formed based on the similarity of credit accounts (a subset of features in this dataset). The task is to categorize clients as good or bad credit risks, with gender as the sensitive attribute. We treat "YearsAtCurrentJob" and "JobClassIsSkilled" as $Z$ (variables other than $S$ that directly affect $X$), and the remaining features as $X$.

For both real-world datasets, the Graph Independence condition (Condition 2) holds because the graph structure is constructed using a separate subset of features, ensuring that it is independent of the exogenous variable $U_X$ associated with the features used for prediction.

### C.2    Implementation

We use a one-layer message-passing neural network to aggregate the sensitive causal effect of one-hop neighbors. We train the MPNN with a learning rate of 0.01 for 500 epochs. All models are implemented using PyTorch 1.12.0 and PyG 2.4.0 and evaluated on a Linux server with an Intel(R) Core(TM) i9-10900X CPU and an NVIDIA GeForce RTX 3070 GPU. The memory consumption is approximately 2000 MiB. We use the cVAE to reconstruct features conditional on $\hat{a}$ and $c$. For cVAE training, the learning rate is 0.01, and the number of epochs is 800. Experimental results are averaged over five repeated executions. We use the Adam optimizer for both components of our proposed framework and implement our method with PyTorch. For the constraint-based method, we train a multilayer perceptron (MLP) with corresponding fairness regularization terms to achieve fairness.

**Risk difference on IID data:** Risk difference refers to the difference in positive prediction rates between the favorable group and the unfavorable group. The probability of an output given a sensitive attribute is $P(y \mid s) = \mathbb{E}_{x|s}P(y \mid x)$. We then define the regularization term as $P(y \mid s^+) - P(y \mid s^-)$.

**Causal inference on IID data:** For IID data, we use a structural causal model to describe the causal relationship between two variables. For example, the causal relationship between $S$ and $X$ is given by:

$$x = f(s, u).$$

We consider the same causal structure in Figure 6 but neglect the network causal effect. To compute the probability of an output under an intervention on the sensitive attribute, $P(y|do(s))$, we use

$$P(y \mid do(s)) = \sum_{z,x} P(z)P(x \mid s, z)P(y \mid x)$$

$$= \sum_{z,x} P(z \mid s)\frac{P(s)}{P(s \mid z)}P(x \mid s, z)P(y \mid x)$$

$$= \sum_{z,x} P(z, x \mid s)\frac{P(s)}{P(s \mid z)}P(y \mid x)$$

$$= \mathbb{E}_{z,x \sim P(z,x|s)}\left[\frac{P(s)}{P(s \mid z)}P(y|x)\right] \tag{6}$$

We then define the regularization term as $P(y \mid do(s^+)) - P(y \mid do(s^-))$.

## C.3 Computational Cost

To compare computational costs, we measured the runtime and memory usage of our method and the baselines on the Credit dataset. As shown in Table 5, MPVA achieves the lowest memory usage (1365 MB) and has the second-fastest runtime (29.2 seconds), after NIFTY (17.0893 seconds), demonstrating its computational efficiency. The superior memory efficiency of MPVA makes it particularly suitable for large-scale graph applications where memory constraints are critical. Notably, GEAR exceeded the 3600-second time limit and could not complete the experiment, highlighting the scalability challenges of existing methods.

Table 5: Runtime and memory usage comparison of different methods.

| Method | Time (s) | Memory (MB) |
|--------|----------|-------------|
| FairGNN | 81.7968 | 3639 |
| NIFTY | 17.0893 | 2419 |
| GEAR | > 3600 | − |
| FairINV | 63.7923 | 2103 |
| MPVA | 29.2358 | 1365 |

