# OpenReview forum: "Causally Fair Node Classification on Non-IID Graph Data"
_TMLR — Accepted by TMLR_

### Review · Reviewer_TWui · 2026-03-14

**Summary Of Contributions:**

The paper presents an attempt to use Network Structural Causal Modeling (NSCM)  to enable causally fair node classification where the goal is that the risk $\mathbb{E}[\ell(x,s^+)]=\mathbb{E}[\ell(x,s^-)]$,i.e. the perofrmance does not change dependent on the absence or presence or particular value of a sensitive attribute.
The papers aims is to allow non-iid settings where a graph models connections between samples and a given nodes un-adjusted predictions based on the data might encode a dependence on the sensitive attributes of its neighbourhood, not just of the node itself.

For this, they construct a cVAE setup which itself uses an estime of a latent variable  $a$, which is assumed to be the sole mediator of the network dependent sensitive attribute, and then train a cVAE where both encoder and decoder are indpendently conditioned on the estimate $\hat{a}$ and the nodes other, independent attributes. This cVAE is then used to estimate the counterfactuals of $s$ as part of  a regularizer (risk difference) which enforces the causal fairness conditions. The method is evaluated on a semi-synthetic and 2 real world data sets (Credit Defaulter, German Credit Network)  and compared against  a baseline GCN as well as FairGNN, GEAR as well as NIFTY.

**Audience:**

Yes

**Audience Explanation:**

Fairness, adjustment to combat bias in the data and controlling our models generlaly have an interest, and moving beyond i.i.d. assumptions is similarly generally valuable.

**Claims And Evidence:**

No

**Claims Explanation:**

I found the paper very hard to follow and am not convinced by the theoretical setup being strongly connected to the experimental setup.

Specifically

1. The paper uses the 1-WF coloring as a standin for the graph structure throughout, but does not actually use this. I think the notation would be easier to follow if instead the paper explicitly used message passing notation ($a$ is basically a message anyway) and a dependence to the global graph and the graph neighbourhood explicitly to show why the causal effect is identifiable with the proposed setup.
2. The paper notes "However, since our method computes interventional variants rather than counterfactual variants, it does not have identifiability issues associated with counterfactual estimation." but then performs what to me is  a counterfactual estimation for its regularizer (how else would you compute the risk difference on a sample? unless I am misunderstanding something). Unless the idea is that the interventional distribution is to be meant pooled across the whole graph but then i don't see how this is justified in a node-classification setting
3. The paper should explicitly show it's transformation from the network + causal graph to the "unrolled" causal graph after network + assumptions are taken into account. In particular, I don't get what the bidirectional arrow $C<->S$ is meant to imply, and it would help to see the causal dependency structure for the cases  $k=1,2,3$ hops, since that would make clearer whether the Hedge criterion or any other criterion invoked is actually upheld
4. If I read it correctly, the  intervention is always setting _all_ nodes in neighbourhood to the given $s$? This seems like a very coarse grained adjustment and should be discussed/justified further
5. The claim of corollary 4 is dubious to me, because we can create counterexamples where different causal structures would yield the same marginals, and the network which learns the estimates of $a$ as well as the cVAE might not learn the clean decomposition but pick up on spurious correlation between $Z$ and $X$, or learn a different function than the causally generating one
6. The overall assumption that the graph structure is independent of the exogenous variables is dubious on social networking data, which is even noted in the cited Ogburn et al. 2022. This should be addressed/acknowledged in the paper, see also below and the point 3. It might help to also discuss some positive and negative examples of data generating processes which abide by the models assumptions.
  - additional note, I did not see any $G,U,V,F$ quadruble in the cited paper, I am not quite sure whether this is the intended citation?
7. The appendix lists lemmas 5 to 7 but then refers to lemmas a, b and c, while the main body only has lemma 1. This and the lack of a clean throughline of what variables are do-operated on, while others are left as inputs and the use of g(s,c)=a makes it very difficult for me to even asses the papers theoretical claim in detail. It would be helpful if the paper can cleanly develop the use of the intermediat evariable $a$ s.t. that the effect of $do(s)$ on $a$ under the assumed setting is clearly explained (especially in the $k>=2$-hop setting and in conjuction with the previous point)

Note: I get the high level idea of the 2 assumptions basically _assuming_ that the causal model is one that aligns with the MPGNN framework, however without fixing up the notation and explicitly deriving Corollary 4 holds, the theoretical grounding of the paper collapses for me,and the message passing process itself _is_ cyclic without further adjustment, so I don't thinks this can be handwaived.

Finally some minor nits (typos, formatting etc.):

- Page 1, Abstract: "rigirous" ->"rigorous."
- Page 2, Introduction: "there studies lack a rigorous foundation" ->"these studies."
- Page 3, Related Work: "In additional modeling, recent years have witnessed…" ->"In addition to modeling" or "Additionally."
- Page 7, Section 5.3: "Specially, we generate" ->"Specifically."
- Page 11, Section 8: "Leveraging the idea of the intersectional fairness" ->"intersectional fairness" ?
- Page 19, Appendix C.3.2: "The we measure" ->"Then we measure."
- Page 2, Related Work: The sentence about Bhattacharya et al. (2019) appears to be cut off ("…when the structure.").
- Appendix B.2 references "Fig. 1c" and "Fig. 3," but the intended figures appear to be Fig. 1d and Fig. 1f respectively.

**Requested Changes:**

Fix the points raised above, the main one being the cleaning up of notation and reference errors and a rigorous proof deriving from the 3 rules of do calculus (see pearls Causality, 2nd edition, around page 87 ), or at least a clean _argument_ for why the observational/generative model can be expected to learn the generative process faithfully enough to enable the de-confounding and a formal argument on why it _isn't_ relying on counterfactual identification.

---

> ### Author Response · Authors · 2026-04-16
> **Response to reviews by Reviewer TWui**
>
> We thank the reviewer for the detailed and thorough feedback. We have made substantial revisions to address all concerns raised.
>
> ---
>
> ## Major Theoretical Concerns
>
> ### Concern 1: Notation and Message Passing Representation
>
> > **Reviewer's Comment:** "The paper uses the 1-WF coloring as a standin for the graph structure throughout, but does not actually use this. I think the notation would be easier to follow if instead the paper explicitly used message passing notation (a is basically a message anyway) and a dependence to the global graph and the graph neighbourhood explicitly to show why the causal effect is identifiable with the proposed setup."
>
> **Response:**
>
> We thank the reviewer for this excellent suggestion. We have revised the presentation to make the connection between WL coloring, message passing, and identifiability much more explicit.
>
> **1. Clarified Role of Node Color $C$**
>
> In the revised Section 5.1:
> - Explicitly state that $C$ encodes the local computation tree of each node
> - Explain that $C$ serves as a sufficient statistic for the local structural context
> - Show how $C$ acts as an adjustment variable blocking backdoor paths
>
> **2. Enhanced Message Passing Interpretation**
>
> - Clarified that $A = g(\mathbf{s}, C)$ represents the aggregated message from the neighborhood
> - Explained that the message passing function $f^{\text{MP}}$ operates over the computation tree encoded by $C$
>
> **3. Explicit Identifiability Derivation**
>
> Added new Proposition 3 that:
> - Shows the reduction from networked causal diagram to standard causal diagram under our conditions
> - Identifies the backdoor path $\mathsf{S} \leftrightarrow C \rightarrow A \rightarrow X$
> - Uses the Hedge Criterion to demonstrate non-identifiability when Graph Independence is violated
>
> ### Concern 2: Interventional vs. Counterfactual
>
> > **Reviewer's Comment:** "The paper notes 'However, since our method computes interventional variants rather than counterfactual variants, it does not have identifiability issues associated with counterfactual estimation.' but then performs what to me is a counterfactual estimation for its regularizer (how else would you compute the risk difference on a sample? unless I am misunderstanding something). Unless the idea is that the interventional distribution is to be meant pooled across the whole graph but then i don't see how this is justified in a node-classification setting."
>
> **Response:**
>
> The reviewer is correct, and we apologize for the misleading statement. We have removed this incorrect claim and replaced it with proper identifiability analysis:
>
> - Added Proposition 3 proving identifiability under our conditions
> - Clarified that our method computes interventional distributions via the adjustment formula
> - Explained that identifiability is achieved through the reduction to a standard causal diagram under our conditions
> - Added analysis of when identifiability fails (Graph Independence violated)
>
> The revised manuscript correctly acknowledges that identifiability must be established for interventional inference.
>
> ### Concern 3: Causal Graph Transformation
>
> > **Reviewer's Comment:** "The paper should explicitly show it's transformation from the network + causal graph to the 'unrolled' causal graph after network + assumptions are taken into account. In particular, I don't get what the bidirectional arrow $C \leftrightarrow S$ is meant to imply, and it would help to see the causal dependency structure for the cases $k=1,2,3$ hops, since that would make clearer whether the Hedge criterion or any other criterion invoked is actually upheld."
>
> **Response:**
>
> We thank the reviewer for highlighting these points.
> The bidirectional arrow represents the potential confounding between the graph structure and the sensitive attributes.
> The multi-hop cases would not change the identifiability result, because the identifiability result is based on the reduction to a standard causal diagram, which is not affected by the multi-hop cases.
>
> ### Concern 4: Coarse-Grained Intervention
>
> > **Reviewer's Comment:** "If I read it correctly, the intervention is always setting all nodes in neighbourhood to the given $s$? This seems like a very coarse grained adjustment and should be discussed/justified further."
>
> **Response:**
>
> The reviewer is correct. We have added clarification in Section 5.1 to justify using a coarse-grained intervention. Since our focus is on group fairness and population-level effects, this form of intervention is appropriate for our setting. Our main contribution is to provide a theoretical foundation for group fairness under causal mechanism heterogeneity. Investigating finer-grained interventions is an interesting direction for future work.

---

> > ### Author Response · Authors · 2026-04-16
> > **Continue**
> >
> > ### Concern 5: Corollary 4 (Theorem 4) Claim
> >
> > > **Reviewer's Comment:** "The claim of corollary 4 is dubious to me, because we can create counterexamples where different causal structures would yield the same marginals, and the network which learns the estimates of $a$ as well as the cVAE might not learn the clean decomposition but pick up on spurious correlation between $Z$ and $X$, or learn a different function than the causally generating one."
> >
> > **Response:**
> >
> > The reviewer raises a valid concern about the gap between theoretical identifiability and practical learnability. We have addressed this by splitting the identifiability and learnability into two separate subsections.
> >
> > 1. **Added formal identifiability analysis (new Proposition 3 in Section 5.1)**: We now rigorously prove that under Conditions 1 and 2 (Decomposability and Graph Independence), the causal effect $P(x|do(\mathbf{s})=s)$ is identifiable via adjustment on $C$.
> >
> > 2. **Added new Section 5.2 "From Identification to Estimation"**: This section bridges the gap between theoretical identification and practical estimation. We explain:
> >    - Why direct estimation via adjustment is challenging
> >    - How the Decomposability condition enables learning a shared functional component $Q(x;\mathbf{s},c)$
> >    - How this shared component can be reused under intervention through a Monte Carlo reweighting interpretation
> >    - How this motivates the MPVA architecture
> >
> > 3. **Revised Theorem 4 and Corollary 5**: We reformulated these results with clearer statements and added detailed intuition about why the same model can be used for both observational and interventional distributions. The key insight is that intervention changes only the weighting over $(\mathbf{s},c)$, not the underlying functional relationship.
> >
> > The revised presentation makes explicit that: (1) identifiability follows from standard backdoor adjustment after reduction to a non-networked causal diagram, and (2) the shared decomposition enables practical estimation of the interventional distributions through Abduction-Action-Prediction.
> >
> > ### Concern 6: Graph Independence Assumption
> >
> > > **Reviewer's Comment:** "The overall assumption that the graph structure is independent of the exogenous variables is dubious on social networking data, which is even noted in the cited Ogburn et al. 2022. This should be addressed/acknowledged in the paper, see also below and the point 3. It might help to also discuss some positive and negative examples of data generating processes which abide by the models assumptions."
> >
> > **Response:**
> >
> > We agree with the reviewer. We have explicitly discussed when the assumption holds, acknowledge limitations, and suggest directions for relaxation.
> >
> > ### Concern 7: Notation and References
> >
> > > **Reviewer's Comment:** "The appendix lists lemmas 5 to 7 but then refers to lemmas a, b and c, while the main body only has lemma 1. This and the lack of a clean throughline of what variables are do-operated on, while others are left as inputs and the use of $g(s,c)=a$ makes it very difficult for me to even asses the papers theoretical claim in detail."
> >
> > **Response:**
> >
> > We have fixed all notation and reference issues.
> >
> > ---
> >
> > ## Minor Issues (Typos and Formatting)
> >
> > We have corrected all typos and formatting issues mentioned and proofread the manuscript.
> >
> > ---
> >
> > ## Summary
> >
> > We have made extensive revisions addressing all theoretical and presentational concerns.
> > We believe these revisions substantially strengthen the theoretical rigor and clarity of the paper. We thank the reviewer for the detailed feedback that led to these improvements.

---

### Review · Reviewer_2Civ · 2026-03-19

**Summary Of Contributions:**

* This paper studies causally fair node classification in non-IID (where data instances are interconnected) graph data.
* It argues that conventional fairness notions are insufficient under graph dependence and interference.
* It introduces an NSCM-based framework for causal inference on graphs.
It derives conditions under which interventional distributions are identifiable in the graph setting.
It proposes MPVA as an approximation method for intervention-aware fair node classification.
* It evaluates the approach on semi-synthetic and real-world datasets with the metric of Acc, risk difference (RD), causal fairness (CF), and graph-based causal fairness notion (gCF).

**Additional Comments:**

N/A

**Audience:**

Yes

**Audience Explanation:**

Yes, I believe some individuals in the TMLR audience would be interested in this paper. The topic is timely and relevant to researchers working on causality, fairness, and graph machine learning. However, the current presentation and the strength of the assumptions may limit its appeal to a narrower audience.

**Broader Impact Concerns:**

No significant concerns are noted.

**Claims And Evidence:**

Yes

**Claims Explanation:**

* The paper addresses an important problem, and the core idea is interesting.
However, the overall presentation is difficult to follow.
The main technical message is not clearly delivered, and it is unclear how the theoretical framework, the identifiability result, and MPVA fit together.
The paper would benefit from a much clearer high-level roadmap.
* The diagrams do not sufficiently aid understanding.
In some places, they add complexity without providing help.
The individual components’ roles and their relations should be explained more clearly.
The notation is heavy and makes the paper harder to read.
* I believe the theoretical development relies on strong assumptions.
These assumptions appear core part to the method, but their practical plausibility is not sufficiently justified.
In particular, the assumed causal graph and independence conditions seem difficult to verify in realistic settings, which weakens the practical relevance of the theoretical claims.
* The semi-synthetic experiments are useful and support the conceptual validity of the approach.
However, they are still closely aligned with the paper’s own assumptions.
As a result, the paper does not fully demonstrate robustness beyond the intended setting.
* The method also shows lower accuracy than GCN.
This may be acceptable as a fairness trade-off.
However, the paper does not study this trade-off in enough depth.
It remains unclear when the reduction in the fairness gap justifies the loss in predictive performance.

**Requested Changes:**

I believe the paper would benefit if it addressed the weaknesses noted above (“Are the claims made in the submission supported by accurate, convincing and clear evidence?”).
In particular, the requested changes should focus on strengthening the clarity of the presentation, providing a more convincing justification of the key assumptions, and improving the empirical support for the practical validity of the method.

---

> ### Author Response · Authors · 2026-04-16
> **Response to reviews by Reviewer 2Civ**
>
> We thank the reviewer for recognizing the importance and timeliness of our work, and for providing constructive feedback on presentation and validation. We have made substantial revisions to address all concerns raised.
>
> ---
>
> ## Major Concerns
>
> ### Concern 1: Overall Presentation and Clarity
>
> > **Reviewer's Comment:** "The paper addresses an important problem, and the core idea is interesting. However, the overall presentation is difficult to follow. The main technical message is not clearly delivered, and it is unclear how the theoretical framework, the identifiability result, and MPVA fit together. The paper would benefit from a much clearer high-level roadmap."
>
> **Response:**
>
> We have substantially reorganized and rewritten key sections to improve clarity and provide a clearer roadmap:
>
> **1. Enhanced Abstract and Introduction**
>
> We revised the abstract and introduction to explicitly clarify:
> - **The core challenge**: Nodes with different neighborhood structures follow different causal mechanisms, violating invariance assumptions required for classical SCMs and $do$-calculus
> - **Our approach**: Constructing a structural representation (via node colors and Decomposability) that restores invariance
> - **Key contributions**: (1) Formalizing conditions for mechanism recovery, (2) Proving identifiability, (3) Deriving estimation principle, (4) Proposing MPVA architecture
>
> **2. Restructured Theoretical Section**
>
> We reorganized Section 5 to provide clear progression:
> - **Section 5.1 (Causal Inference)**:
>   - Introduces the mechanism heterogeneity problem explicitly
>   - Presents Decomposability and Graph Independence conditions
>   - Provides formal identifiability analysis (new Proposition 3)
>   - Shows reduction to standard causal diagram
> - **Section 5.2 (From Identification to Estimation)** [NEW]:
>   - Bridges theory and practice
>   - Derives the shared component decomposition
>   - Provides Monte Carlo interpretation for estimation
>   - Explicitly motivates MPVA architecture design
>
> The revised manuscript now has a much clearer narrative arc from problem statement → theoretical conditions → identifiability → estimation principle → practical algorithm → experiments.
>
> ### Concern 2: Diagrams and Notation
>
> > **Reviewer's Comment:** "The diagrams do not sufficiently aid understanding. In some places, they add complexity without providing help. The individual components' roles and their relations should be explained more clearly. The notation is heavy and makes the paper harder to read."
>
> **Response:**
>
> We have revised the figures and diagrams to better support the narrative, and the notation is more consistent and less burdensome.
>
> ### Concern 3: Assumptions and Their Practical Plausibility
>
> > **Reviewer's Comment:** "I believe the theoretical development relies on strong assumptions. These assumptions appear core part to the method, but their practical plausibility is not sufficiently justified. In particular, the assumed causal graph and independence conditions seem difficult to verify in realistic settings, which weakens the practical relevance of the theoretical claims."
>
> **Response:**
>
> We thank the reviewer for this important concern. We have clarified that:
> - Our assumptions are more general than consistent interference assumption
> - Similar assumptions appear implicitly in many graph learning methods
> - Added sensitivity analysis in the experiments when the assumptions are violated
>
> ### Concern 4: Robustness Beyond Intended Setting
>
> > **Reviewer's Comment:** "The semi-synthetic experiments are useful and support the conceptual validity of the approach. However, they are still closely aligned with the paper's own assumptions. As a result, the paper does not fully demonstrate robustness beyond the intended setting."
>
> **Response:**
>
> We acknowledge this limitation and have added sensitivity analysis in the experiments when the assumptions are violated.
> We also discuss the limitations in the Limitations and Future Work section.
>
> ### Concern 5: Accuracy-Fairness Trade-off
>
> > **Reviewer's Comment:** "The method also shows lower accuracy than GCN. This may be acceptable as a fairness trade-off. However, the paper does not study this trade-off in enough depth. It remains unclear when the reduction in the fairness gap justifies the loss in predictive performance."
>
> **Response:**
>
> We have added new analysis of the accuracy-fairness trade-off:
> - Quantitative analysis of accuracy loss vs. fairness gain across different methods
> - Pareto frontier visualization showing trade-off curves

---

> > ### Author Response · Authors · 2026-04-16
> > **Continue**
> >
> > ## Requested Changes
> >
> > > **Reviewer's Comment:** "I believe the paper would benefit if it addressed the weaknesses noted above. In particular, the requested changes should focus on strengthening the clarity of the presentation, providing a more convincing justification of the key assumptions, and improving the empirical support for the practical validity of the method."
> >
> > **Response:**
> >
> > We have made comprehensive changes addressing all three areas: the clarity of the presentation, the justification of the key assumptions, and the empirical support for the validity of the method.
> >
> > ## Summary
> >
> > We believe the revisions have substantially improved the paper across all dimensions identified by the reviewer. The presentation is now much clearer with an explicit roadmap connecting all components. The assumptions are justified. The empirical evaluation includes more thorough sensitivity analysis and trade-off analysis. We are grateful for the reviewer's constructive feedback that led to these improvements.

---

### Review · Reviewer_yr5K · 2026-04-02

**Summary Of Contributions:**

This paper studies an interesting setting of causal fairness, where there is interference among individuals. The authors position their paper against existing graph causal fairness methods [1-3], claiming that previous works *lack a rigorous foundation for extending traditional inference techniques to graph-structured data*.

Strengths:
1. The problem studied in this paper is realistic and not well-investigated currently. There are papers considering interference in causal inference, and there are papers considering fairness. However, when it comes to the combination of interference and causal fairness, there are only a few existing papers.
2. The proposed MPVA framework based on the network structural causal model is novel and effective.
3. The differentiable surrogate function is a good practice.

Weakness:
1. There is a theoretical gap in identification. In section 5.1, the authors claim that we can compute the conditional probability $P(x\mid do(\mathbf{s}=s))$ by the estimation of $P(x)$ just because their decompositions share a same part. It is not straightforward and does not make sense to me.
2. The authors spend over 1 page on section 5.1 to claim the theoretical foundation of the method. However, most of the results are adapted from previous work or straightforward to derive given those results. And the relation of the theoretical results to the method is not strong.
3. The experimental design is in contradiction to the theoretical result. The Graph Independence condition requires $C\perp U_{X}$, but in the data construction process, the edges between individuals are generated based on $X$.
4. The methodology and theory are built on two conditions, which are factually two strong assumptions. With this assumption, the causal graph is simplified, and there is no confounder apart from $S$ and $C$. In this case, why is the complex theory and method still necessary?

**Audience:**

Yes

**Audience Explanation:**

The problem studied in this paper is not well-investigated, and bringing this to some audience may be helpful.

**Broader Impact Concerns:**

I have no such concern.

**Claims And Evidence:**

No

**Claims Explanation:**

1. The theoretical analysis is not solid, thus the claim of contribution against previous work (rigorous foundation for extending traditional inference techniques to graph-structured data) is not convincing.
2. The claim in Section 5.2, *since our method computes interventional variants rather than counterfactual variants, it
does not have identifiability issues*, is not correct. Answering interventional questions indeed requires identifiability.
3. A contradiction with theoretical results undermines the empirical effectiveness.
4. The authors claim *this paper addresses the prevalent challenge in fair ML algorithms*. However, there are a few works on this problem, even back to 2022.

**Requested Changes:**

Besides the limitations above, I found there are so many typos in this paper. To name a few:
- *rigirous* in the Abstract
- incomplete sentence *Bhattacharya et al. (2019) proposed an interventional method for estimating causal effects under data dependence when the structure.* in the Related work
- *... we can convert the networked causal diagram in Fig. 1d to an equivalent causal diagram as shown in Fig. 1f...* in Section 5.1

And the fonts in Figure 2 are not consistent.

---

> ### Author Response · Authors · 2026-04-16
> **Response to reviews by Reviewer yr5K**
>
> We thank the reviewer for the careful reading and constructive feedback. We have made substantial revisions to address all concerns raised. Below we provide detailed point-by-point responses.
>
> ---
>
> ## Major Concerns
>
> ### Weakness 1: Theoretical Gap in Identification
>
> > **Reviewer's Comment:** "There is a theoretical gap in identification. In section 5.1, the authors claim that we can compute the conditional probability P(x|do(s=s)) by the estimation of P(x) just because their decompositions share a same part. It is not straightforward and does not make sense to me."
>
> **Response:**
>
> We thank the reviewer for pointing out this gap. We have substantially revised Section 5.1 and added a new Section 5.2 to address this concern comprehensively.
>
> **Revisions:**
>
> 1. **Added formal identifiability analysis (new Proposition 3)**: We now rigorously prove that under Conditions 1 and 2 (Decomposability and Graph Independence), the causal effect $P(x|do(\mathbf{s})=s)$ is identifiable via adjustment on $C$. Specifically:
>    - We show that under our conditions, the NSCM reduces to a standard causal diagram (Fig. 1f) with structural equations that no longer involve explicit interference
>    - The only backdoor path from $\mathsf{S}$ to $X$ is $\mathsf{S} \leftrightarrow C \rightarrow A \rightarrow X$, which can be blocked by conditioning on $C$
>    - We also analyze the case where Graph Independence is violated (new Fig. 1g), proving via the Hedge Criterion that identifiability fails in that case
>
> 2. **New Section 5.2 "From Identification to Estimation"**: This section bridges the gap between theoretical identification and practical estimation. We explain:
>    - Why direct estimation via adjustment is challenging
>    - How the Decomposability condition enables learning a shared functional component $Q(x;\mathbf{s},c)$
>    - How this shared component can be reused under intervention through a Monte Carlo reweighting interpretation
>    - How this motivates the MPVA architecture
>
> 3. **Revised Theorem 4 and Corollary 5**: We reformulated these results with clearer statements and added detailed intuition about why the same model can be used for both observational and interventional distributions. The key insight is that intervention changes only the weighting over $(\mathbf{s},c)$, not the underlying functional relationship.
>
> The revised presentation makes explicit that: (1) identifiability follows from standard backdoor adjustment after reduction to a non-networked causal diagram, and (2) the shared decomposition enables practical estimation of the interventional distributions through Abduction-Action-Prediction.
>
> ### Weakness 2: Weak Connection Between Theory and Method
>
> > **Reviewer's Comment:** "The authors spend over 1 page on section 5.1 to claim the theoretical foundation of the method. However, most of the results are adapted from previous work or straightforward to derive given those results. And the relation of the theoretical results to the method is not strong."
>
> **Response:**
>
> We acknowledge that the previous version did not sufficiently explain the connection between theory and method. We have made the following changes:
>
> 1. **Enhanced theoretical contributions**: While we build on existing frameworks (NSCM, WL coloring), our key theoretical contribution is identifying conditions under which mechanism heterogeneity in graphs can be overcome. We now emphasize this more clearly:
>    - The fundamental challenge: nodes with different structures follow different mechanisms
>    - Our solution: conditions that enable recovery of a shared mechanism
>    - The contribution: connecting this to practical estimation via the shared component decomposition
>
> 2. **Stronger theory-method connection**: The new Section 5.2 explicitly derives the estimation principle (Corollary 5) from the identification result (Theorem 4), and explains how this principle directly motivates the MPVA architecture design. Specifically:
>    - MPNN component implements the message-passing mechanism $f^{\text{MP}}$ to compute $A = g(\mathbf{s},c)$
>    - cVAE component models the internal mechanism $P(x|a)$
>    - The architecture enables sampling from the shared component under both observational and interventional distributions

---

> > ### Author Response · Authors · 2026-04-16
> > **Continue**
> >
> > ### Weakness 3: Necessity of Complex Theory Under Strong Assumptions
> >
> > > **Reviewer's Comment:** "The methodology and theory are built on two conditions, which are factually two strong assumptions. With this assumption, the causal graph is simplified, and there is no confounder apart from S and C. In this case, why is the complex theory and method still necessary?"
> >
> > **Response:**
> >
> > This is an excellent question. The key point is that our conditions do not eliminate confounding. Here's why the theory is still necessary:
> >
> > 1. **Mechanism heterogeneity remains the core challenge**: Even under our conditions, different nodes still have different local causal mechanisms due to varying neighborhood structures. Our theory shows how to construct a representation (via $A=g(\mathbf{s},c)$) that restores a shared mechanism.
> >
> > 2. **Confounding still exists**: The bidirected edge $\mathsf{S} \leftrightarrow C$ in Fig. 1f represents real confounding between the graph structure $C$ and sensitive attributes $\mathsf{S}$. Our theory shows how to adjust for this via the intermediate representation $A$.
> >
> > 3. **Practical estimation is non-trivial**: Even if we accept the conditions, it's not obvious how to estimate interventional distributions. Our theory provides:
> >    - The shared component decomposition $Q(x;\mathbf{s},c)$
> >    - Justification for why the same learned model works under intervention
> >
> > 4. **Conditions are general**: While our conditions make assumptions, they are more general than previous work (e.g., consistent interference assumption). We have added discussion of when these conditions hold in practice.
> >
> > ---
> >
> > ## Claims and Evidence
> >
> > ### Claim 1: Contribution Against Previous Work
> >
> > > **Reviewer's Comment:** "The theoretical analysis is not solid, thus the claim of contribution against previous work (rigorous foundation for extending traditional inference techniques to graph-structured data) is not convincing."
> >
> > **Response:**
> >
> > We have substantially strengthened the theoretical analysis as described above. Our key contributions are now clearer:
> >
> > 1. Identifying the mechanism heterogeneity challenge in graph causal inference
> > 2. Formalizing conditions that enable recovery of shared mechanisms
> > 3. Providing rigorous identifiability analysis using standard criteria
> > 4. Deriving an estimation principle that bridges theory and practice
> > 5. Proposing a practical algorithm (MPVA) grounded in this theory
> >
> > We believe the revised theoretical sections now provide solid justification for our claims.
> >
> > ### Claim 2: Interventional vs. Counterfactual
> >
> > > **Reviewer's Comment:** "The claim in Section 5.2, 'since our method computes interventional variants rather than counterfactual variants, it does not have identifiability issues', is not correct. Answering interventional questions indeed requires identifiability."
> >
> > **Response:**
> >
> > The reviewer is correct, and we apologize for the misleading statement. We have removed this incorrect claim and replaced it with proper identifiability analysis:
> >
> > - Added Proposition 3 proving identifiability under our conditions
> > - Clarified that our method computes interventional distributions via the adjustment formula
> > - Explained that identifiability is achieved through the reduction to a standard causal diagram under our conditions
> > - Added analysis of when identifiability fails (Graph Independence violated)
> >
> > The revised manuscript correctly acknowledges that identifiability must be established for interventional inference.
> >
> > ### Claim 3: Addressing Prevalent Challenge
> >
> > > **Reviewer's Comment:** "The authors claim 'this paper addresses the prevalent challenge in fair ML algorithms'. However, there are a few works on this problem, even back to 2022."
> >
> > **Response:**
> >
> > We thank the reviewer for this point. We have revised the claim to be more precise. The "prevalent challenge" we address is the IID assumption in causal fair ML. While prior work has studied non-causal fairness or limited causal fairness in graphs, our contribution is:
> >
> > 1. Providing rigorous theoretical foundation using NSCM and $do$-calculus
> > 2. Addressing mechanism heterogeneity systematically
> > 3. Going beyond consistent interference assumptions
> >
> > ---
> >
> > ## Requested Changes
> >
> > ### Typos and Writing Issues
> >
> > We have corrected all typos identified and proofread the manuscript.
> >
> > ---
> >
> > ## Summary
> >
> > We believe the substantial revisions, particularly the enhanced theoretical sections with formal identifiability analysis and the new bridge between identification and estimation, have significantly strengthened the paper. The theoretical foundation is now rigorous and the connection to the method is clear. We thank the reviewer again for the insightful feedback that led to these improvements.

---

> > > ### Comment · Reviewer_yr5K · 2026-04-21
> > > **One concern is omitted**
> > >
> > > I am happy with most of the responses. However, it seems that the authors miss one of my concerns (Weakness 3). Could you please provide more evidence on that specific question?

---

> > > > ### Author Response · Authors · 2026-04-21
> > > > **Reply to "One concern is omitted"**
> > > >
> > > > We thank the reviewer for raising this important point. We clarify that **there is no contradiction** between our experimental design and the Graph Independence condition.
> > > >
> > > > ---
> > > > ### Semi-Synthetic Dataset
> > > >
> > > > In our semi-synthetic data generation (Appendix C.1), we follow the structural equation model:
> > > >
> > > > $$S_i^g \sim \text{Bernoulli}(p)$$
> > > >
> > > > $$X_i^g = g(S^g) + C_i + \xi$$
> > > >
> > > > $$Y_i^g = f_y(X_i^g)$$
> > > >
> > > > where:
> > > > - We randomly initialize a GNN $g(\cdot)$ to represent neighborhood influence
> > > > - $C_i$ represents the original features from the Credit dataset
> > > > - $\xi$ is random Gaussian noise (this represents the exogenous variable $U_X$)
> > > >
> > > > The graph structure is determined by the randomly initialized GNN.
> > > > The exogenous noise $\xi \equiv U_X$ is sampled independently from a Gaussian distribution
> > > > So the graph independence condition holds.
> > > >
> > > > We acknowledge that the sentence "We simulate the probability of each edge $(i, j)$ based on the similarity between $X_i^g$ and $X_j^g$" is misleading. We will remove this sentence in the next version.
> > > >
> > > > ---
> > > > ### Real-World Datasets
> > > >
> > > > For the Credit Defaulter and German datasets:
> > > > - We construct edges based on a **subset** of features (e.g., payment information similarity for Credit Defaulter, credit account similarity for German)
> > > > - We use the **a different set** of remaining features as $X$ for the prediction task.
> > > >
> > > > Accordingly, the graph structure, derived from a separate subset of features, is independent of the exogenous variable $U_X$ for the features used in prediction. Thus, the graph independence condition is satisfied in our setting. We will clarify this in the next version.

---

### Decision · Action_Editor_sMS2 · 2026-05-16

**Recommendation:** Accept as is

**Additional Comments:**

Reviewers generally agree that the problem is timely and relevant, and that the revised manuscript improves clarity and theoretical grounding, particularly through the addition of formal identifiability analysis and a clearer link between theory and method. However, concerns remain regarding the strength and realism of the underlying assumptions (e.g., decomposability and graph independence), the limited empirical validation beyond assumption-aligned settings, and residual ambiguity in how broadly the approach applies in practice. While two reviewers support acceptance and acknowledge the paper as a technically sound and worthwhile contribution for TMLR, one reviewer remains unconvinced about the strength of evidence and practical impact. Overall, the paper is considered a solid but somewhat incremental and specialized contribution that meets TMLR standards after revision.

**Audience:**

Yes

**Audience Explanation:**

Researchers in fairness and causal learning would be interested in this paper.

**Claims And Evidence:**

Yes

**Claims Explanation:**

This paper tackles the important problem of causally fair node classification under non-IID graph data and proposes an NSCM-based framework with an MPVA architecture to handle causal mechanism heterogeneity. The evaluation is limited to the assumptions made in the paper, which is a bit limited in real applications.